# Amyloid-Forming Corpora Amylacea and Spheroid-Type Amyloid Deposition: Comprehensive Analysis Using Immunohistochemistry, Proteomics, and a Literature Review

**DOI:** 10.3390/ijms25074040

**Published:** 2024-04-04

**Authors:** Shojiro Ichimata, Yukiko Hata, Tsuneaki Yoshinaga, Nagaaki Katoh, Fuyuki Kametani, Masahide Yazaki, Yoshiki Sekijima, Naoki Nishida

**Affiliations:** 1Department of Legal Medicine, Faculty of Medicine, University of Toyama, Toyama 930-0194, Japan; 2Department of Medicine (Neurology and Rheumatology), Shinshu University School of Medicine, Matsumoto 390-8621, Japan; kiccho828@gmail.com (T.Y.);; 3Department of Brain and Neurosciences, Tokyo Metropolitan Institute of Medical Science, Tokyo 156-8506, Japan; kametani-fy@igakuken.or.jp; 4Institute for Biomedical Sciences, Shinshu University, Matsumoto 390-8621, Japan; mayazaki@shinshu-u.ac.jp

**Keywords:** amyloid, beta-2 micro-globulin, corpora amylacea, functional amyloids, lactoferrin, lysosome, macrophage, pulmonary surfactant protein

## Abstract

This study aimed to elucidate the similarities and differences between amyloid-forming corpora amylacea (CA) in the prostate and lung, examine the nature of CAs in cystic tumors of the atrioventricular node (CTAVN), and clarify the distinctions between amyloid-forming CA and spheroid-type amyloid deposition. We conducted proteomics analyses using liquid chromatography–tandem mass spectrometry with laser microdissection and immunohistochemistry to validate the characteristics of CAs in the lung and prostate. Our findings revealed that the CAs in these organs primarily consisted of common proteins (β2-microglobulin and lysozyme) and locally produced proteins. Moreover, we observed a discrepancy between the histopathological and proteomic analysis results in CTAVN-associated CAs. In addition, while the histopathological appearance of the amyloid-forming CAs and spheroid-type amyloid deposits were nearly identical, the latter deposition lacked β2-microglobulin and lysozyme and exhibited evident destruction of the surrounding tissue. A literature review further supported these findings. These results suggest that amyloid-forming CAs in the lung and prostate are formed through a shared mechanism, serving as waste containers (wasteosomes) and/or storage for excess proteins (functional amyloids). In contrast, we hypothesize that while amyloid-forming CA and spheroid-type amyloid deposits are formed, in part, through common mechanisms, the latter are pathological.

## 1. Introduction

Corpora amylacea (CA) has been described in various human organs, including the brain, lungs, and prostate [1,2,3,4]. CA primarily involves two different types of lesions: one is a polyglucosan structure similar to starch, which is positive by periodic acid–Schiff staining, whereas the other is an aggregate of certain types of fibrillary proteins that stain positively for Congo red (CR) and thioflavin [1]. Although these are histopathologically distinct lesions, they are believed to share a common function as wastesomes [1]. Amyloid-forming CA formation is frequently observed in the prostate and lungs [1,5,6]. Several histopathological and biochemical studies have evaluated the CA of the prostate [7,8], but few studies have examined CA in the lung. Furthermore, there have been no studies examining the similarities and differences between CAs at the two sites.

Cystic tumor of the atrioventricular node (CTAVN) is a rare, benign lesion located at the base of the atrial septum, around the area of the AVN in the human heart. CAs are a histopathological feature of CTAVN [4,9]. These deposits may have amyloid properties because of their green birefringence under polarized light [9]; however, there has been no characterization of these deposits, particularly as to whether they contain amyloid-associated proteins, based on proteomic analyses.

Spheroid-type amyloid deposition (STAD) is a rare form of amyloidosis [10,11,12]. This pattern of deposition often appears as a localized form, but it can also be associated with systemic amyloidosis. Morphologically, amyloid-forming CA and STAD are very similar; however, the latter lesions often occur in association with tumors or cause damage to surrounding tissues, which suggests that they are pathologically distinct lesions. To our knowledge, no studies have thoroughly examined the differences between these lesions.

The objectives of this study were as follows: (1) elucidate the similarities and differences between amyloid-forming CAs in the prostate and lung; (2) determine the nature of CAs in CTAVN; and (3) clarify the differences between amyloid-forming CA and STAD. We conducted proteomics analyses using liquid chromatography–tandem mass spectrometry (LC-MS/MS) with laser microdissection (LMD) in a patient with pulmonary and prostatic amyloid-forming CAs. Immunohistochemistry was performed in a series of three patients with pulmonary CA and three patients with prostatic CA to validate the results of the proteomic analyses. Moreover, we performed the same proteomic analysis in a patient with CTAVN-associated CR-positive CA. To address the third objective, we re-evaluated the results of our previously reported histopathologic and proteomic analyses of a STAD patient [10]. Finally, we reviewed prior histopathological studies on STAD and compared their findings with those obtained in the present study on amyloid-forming CA.

## 2. Results

### 2.1. General Appearance of Pulmonary, Prostatic, and CTAVN-Associated CAs

We selected three cases of pulmonary CA, three cases of prostatic CA, three cases of CA associated with CTAVN, and one case of STAD from the archives of all medicolegal autopsy patients from our department between January 2008 and August 2023. A summary of the cases is presented in Table 1.

### 2.2. General Appearance of Pulmonary, Prostatic, and CTAVN-Associated CAs

Representative histopathological microphotographs are shown in Figure 1.

Generally, all the CAs exhibited round, eosinophilic deposits with concentric laminations by H&E staining (Figure 1a–c). The pulmonary CAs exhibited a relatively uniform size and shape, with weak-to-moderate congophilia, and they exhibited clear apple-green birefringence under polarized light observation (Figure 1a,d,g). In contrast, the prostatic CAs varied in size and shape, with strong congophilia and distinct apple-green birefringence under polarized light (Figure 1b,e,h). The CTAVN-associated CAs also varied in size and shape, with weak-to-moderate congophilia, and with only some displaying apple-green birefringence under polarized light (Figure 1c,f,i). In addition, macrophages were observed around the CAs in the lung (Figure 1a) and prostate (Figure 1b) but not around the CTAVN-associated CAs (Figure 1c). There was no evident destruction of the surrounding tissue by the CAs in all cases.

### 2.3. Proteomic and Immunohistochemical Features of Pulmonary CAs

A summary of the proteomics results and representative microphotographs of the pulmonary CAs in Case 1 are shown in Figure 2. Additional microphotographs of the pulmonary CAs in all three cases are provided in Appendix A.

LC-MS/MS with LMD detected five proteins thought to be amyloidogenic, as shown in Figure 2a. Immunohistochemical localization within the CAs was confirmed for β2-microglobulin (Figure 2b), lysozyme C (Figure 2c), pulmonary surfactant protein (PSP)-A, and PSP-B (Figure 2d,e). Immunohistochemistry for lactoferrin yielded negative results (Figure 2f), whereas focal p62 immunoreactivity was observed (Figure 2g).

### 2.4. Proteomic and Immunohistochemical Features of Prostatic CAs

A summary of the proteomics results for the prostatic CAs in Case 1, along with representative microphotographs of Cases 1 and 2, are shown in Figure 3. Representative microphotographs of the prostatic CAs for all three cases are provided in Appendix A.

In addition to some amyloid-associated proteins, three proteins that may be amyloidogenic were detected by LC-MS/MS with LMD (Figure 3a), which is consistent with previous studies [7,8]. Immunohistochemistry revealed that all the CAs were positive for lysozyme C (Figure 3b) and lactoferrin (Figure 3c). Focally, p62 immunoreactivity was identified (Figure 3d). Interestingly, eosinophilic deposits showing moderate congophilia and apple-green birefringence under polarized light were identified within a few carcinoma glands (Figure 3e–g); however, lysozyme C, lactoferrin, and p62 were not detected in these deposits by immunohistochemistry (Figure 3h–j).

### 2.5. Comparison of the Amino Acid Sequences of Proteins Commonly Identified in Pulmonary and Prostatic CAs

The amino acid sequences and identified peptide sequences of β2-microglobulin, lysozyme C, and lactoferrin in the lungs and prostate are shown in Figure 4.

For β2-microglobulin and lysozyme C, nearly identical regions were identified in the lung and prostate (Figure 4a–d); however, for lactoferrin, only a small portion was identified in the lung (Figure 4e), particularly on the C-terminal end, whereas almost the entire length of lactoferrin was identified in the prostate (Figure 4f). The detected PSP-A and PSP-B peptide sequences are shown in Appendix A.

### 2.6. Proteomic and Immunohistochemical Features of CTAVN-Associated CAs

A summary of the proteomics results and representative microphotographs of the immunohistochemistry in Case 1 are shown in Figure 5.

In contrast to the pulmonary and prostatic CAs, we did not detect amyloid-associated or reported amyloidogenic proteins in the CTAVN-associated CAs by LC-MS/MS with LMD (Case 1); however, consistent with our previous report [4], olfactomedin 4 was detected in the CAs (Figure 5a,b). The CAs of all three cases exhibited weak-to-strong immunoreactivity for olfactomedin 4 (Figure 5c–e). In contrast, p62 immunoreactivity was identified only in Case 1 (Figure 5f–h). The representative findings for all three CTAVN cases are shown in Appendix A.

### 2.7. Proteomic and Immunohistochemical Features of STAD Associated with Systemic Amyloid Immunoglobulin Light Chain (AL) Amyloidosis

Representative microphotographs are shown in Figure 6.

In this case, the shape of the amyloid deposits was similar to that of the CAs, particularly in the prostate; however, the destruction of surrounding tissue by the deposits was evident. Furthermore, although some amyloid-associated proteins and seven peptides from the immunoglobulin light chain lambda variable region were detected, β2-microglobulin and lysozyme C were not detected. Furthermore, there was no p62 immunoreactivity in the deposits.

### 2.8. Literature Review of Previous Reports on STAD

To further characterize STAD, we conducted a comprehensive literature review and have summarized the results in Table 2 [10,11,12,14,15,16,17,18,19,20,21,22,23,24,25,26,27,28,29,30,31,32,33,34,35,36,37,38,39,40,41,42,43,44,45].

STAD was observed in both localized and systemic amyloidosis. Although the size and morphology varied between cases and among the organs, the deposits appeared to show concentric laminations in most cases. In addition, the presence of histiocytes and multinucleated giant cells was noted around the deposits in some cases, suggesting that they may be involved in the formation of the deposits [10,11]. These features are similar to those observed in amyloid-forming CAs. Consistent with our results, immunohistochemistry and proteomics analysis did not identify β2-microglobulin or lysozyme C in the deposits, and tissue damage resulting from the deposits was believed to have occurred in almost all STAD cases.

## 3. Discussion

In the present study, we demonstrated that: (1) CAs in the lung and prostate are comprised of common proteins (β2-microglobulin and lysozyme C) and locally produced proteins (PSP-A and PSP-B in the lung; lactoferrin in the prostate); (2) CAs associated with CTAVN exhibit typical histopathological features of amyloid, whereas LMD with LC-MS/MS did not detect amyloid-associated or amyloidogenic proteins; (3) STAD is morphologically similar to amyloid-forming Cas, but it differs because it causes tissue destruction and lacks β2-microglobulin or lysozyme C deposition (4). p62 immunoreactivity was observed in amyloid-forming Cas but not in STAD.

Several proteomic and immunohistochemical analyses of prostatic CA have been reported [7,8,46,47,48]; however, to our knowledge, this is the first comprehensive analysis of pulmonary CA using both methods. CAs in the lung and prostate consist of common proteins (β2-microglobulin and lysozyme C) and locally produced proteins. This suggests a common mechanism underlying the formation of CAs in the lungs and prostate. CAs are frequently identified in normal prostate tissue [49], and their presence does not appear to damage the surrounding tissues in the prostate and lung. Thus, we can speculate that CAs in these organs are formed for physiological reasons rather than serving a pathological function. Riba et al. proposed that CAs act as “wasteosomes” to sequester waste products [1,50]. In contrast, certain polypeptide hormones are stored in an amyloid-like β-sheet conformation in the pituitary gland, which are considered functional amyloids [13,51]. Thus, it has been hypothesized that certain proteins, especially those prone to form amyloid fibrils, are disposed of or stored as amyloid-forming CA. Partial p62 immunoreactivity supports the role of amyloid-forming CAs as wasteosomes. β2-microglobulin and lysozyme C may be essential components for the formation of amyloid CAs.

Macrophages were observed around the CAs in both organs and were also positive for β2-microglobulin and lysozyme C, suggesting that they play an important role in this mechanism. Dobashi et al. suggested that the concentrically laminated bodies in pulmonary CAs may be formed by sequential aggregation, fusion, coalescence, and compaction of degenerated alveolar macrophages [6]; however, it remains unclear whether they remove deposits, create them, or perform both functions. Further studies are needed to elucidate this mechanism, as it may contribute to the development of amyloid removal therapy.

There are four types of PSPs: hydrophilic PSP-A and PSP-D, and hydrophobic PSP-B and PSP-C [52]. In pulmonary CAs, proteomic analysis revealed PAP-A and PAP-B, whereas PSP-C, known to form amyloid fibrils [53], was not detected. These results were confirmed by immunohistochemistry and are consistent with those of a previous report [54]. To our knowledge, however, there are no reports of PAP-A and PSP-B forming amyloid fibrils. Thus, it is unclear whether the PSP-A and PSP-B proteins within the pulmonary CA form amyloid fibrils or are merely deposited. During the metabolism of PSPs, >50% is derived from recycling or catabolic events managed by functional cross-talk between alveolar type II cells and macrophages [52]. We hypothesize that excess or denatured PSPs are converted into CAs to be discarded or stored as a stable structure. It is noteworthy that if this serves a storage function, it may represent a form of functional amyloids [55].

We observed pCR-positive deposits not only in benign acini but also in cancer acini. Although CAs are primarily noted in benign acini, they have been identified in 0.4%–13% of cancer acini [49]. Therefore, the observation of amyloid-forming deposits in the lumen of cancer acini is not surprising; however, it is noteworthy that neither the tumor glands nor their luminal deposits were immunoreactive to lactoferrin. This is consistent with the proteomic results of Tekin et al. [47]. Thus, it is likely that the composition of amyloid deposits in cancer acini differs from those in benign acini, which may lead to the discovery of a novel amyloid precursor protein. However, it should be noted that given the presumed nonphysiological nature of secretions in cancer acini, there may be a discrepancy between the histological findings of the deposits and the results obtained from proteomic analysis, as illustrated in the case of CTAVN, which is discussed below. This is a topic for future consideration.

Interestingly, all CTAVN cases exhibited CAs positive for pCR staining and displayed apple-green birefringence under polarized light; however, proteomic analysis yielded divergent results, with no accumulation of amyloid-associated proteins. The precise reason for this discrepancy remains unknown. One hypothesis is that CTAVN CAs exhibit significant variation in the degree of amyloid fibril formation, given the substantial differences in the congophilia of each deposit. Consequently, it is plausible that only deposits lacking amyloid formation were selected during the LMD process. Another possibility is that congophilia is a false positive. Because CTAVN is a non-physiologic lesion, the CAs forming within the glands of CTAVN may have different properties compared with those in the lung or prostate. The immunohistochemical characteristics of the CTAVN epithelium are similar to those of the prostate epithelium [4]; however, the proteins comprising CA are distinctly different in the two tissues. Moreover, given the significant sex-dependent differences observed in the immunohistochemical properties of the epithelium of CTAVN [4], it is important to consider that the proteomic analysis of CAs from the female patients in this study may have generated results distinct from those of the prostate. It will be necessary to collect more cases of CTAVN and analyze whether amyloid fibrils are indeed formed in CAs associated with CTAVN.

Through morphological examination and a literature review, we demonstrated that amyloid-forming CAs and STADs exhibit morphological similarities; however, the former do not exhibit distinct destruction of the surrounding tissue, in contrast to STAD. Collectively, these findings suggest that the formation of characteristic deposition morphology in amyloid-forming CAs and STAD may be partially mediated by common mechanisms, whereas the former are more closely associated with physiological processes, while the latter is likely formed through pathological mechanisms. This is supported by the observation that p62 was partially positive in the amyloid-forming CAs, whereas it was negative in the STADs. Nevertheless, to our knowledge, there are limited studies showing the immunoreactivity of proteins related to waste substance processing and elimination, such as ubiquitin or p62, in STAD [3,23]. Additional research in this area is warranted.

In addition to a certain level of bias in our study population, this study was also limited by the small sample size, with only three cases with CA being evaluated for each organ. Additionally, potential sex differences were not evaluated given that prostate is a male-specific organ. Furthermore, only one case with STAD was evaluated. Finally, we did not explore structural variations in amyloid fibers in lung and prostate CAs or such differences between CAs and STADs, underscoring the need for further investigations employing cryogenic electron microscopy.

In conclusion, we present the findings of a comprehensive analysis that integrates proteomic analysis and immunohistochemistry for amyloid-forming CAs. A summary of the results is presented in Table 3.

These results suggest that amyloid-forming CAs in the lung and prostate may be formed through a shared mechanism, serving as waste containers (wasteosomes) and/or storage for excess proteins (functional amyloids). It would be interesting if the human body intentionally employs structures that manifest distinctive amyloid fibril formation at the histological level. In contrast, in the CAs associated with CTAVN, there was a discrepancy between the histopathological and proteomic analysis results, highlighting the presence of nonphysiological functions of the epithelium in this disease. Furthermore, while amyloid-forming CA and STAD are formed, in part, by some common mechanisms, the former may have physiological origins, whereas the latter is pathological. If this shared mechanism exists, its elucidation will contribute to the development of amyloid removal therapy via approaches distinct from those using antibody-based methods. Moreover, the incidence of CAs increases with age [1,7], suggesting that age-related shifts in protein production and clearance may influence CA development. Given the association of many amyloidoses with aging, alterations in protein metabolism due to aging might be a critical factor in the underlying pathogenesis. Therefore, we speculate that by analyzing CA and identifying the specific protein metabolic mechanisms that are altered with aging, insights into approaches that can be utilized to prevent amyloid deposition may be gained.

## 4. Materials and Methods

### 4.1. Tissue Samples

Tissue specimens from the lung (for evaluating pulmonary CA), prostate (for evaluating prostatic CA), heart (for evaluating CTAVN-associated CA), and kidney (for evaluating STAD) were collected, fixed in 20% buffered formalin, and routinely embedded in paraffin. Then, 4 μm thick sections were cut and stained with hematoxylin and eosin (H&E) or were analyzed by immunohistochemistry. Furthermore, 6 μm thick sections were cut and stained with phenol CR (pCR) [56].

### 4.2. Histopathological Evaluation

The presence of CAs was determined using H&E-stained specimens. pCR-positive structures, which showed typical apple-green birefringence under polarized light, were histologically confirmed as amyloid deposits. Immunohistochemistry was performed to confirm the proteomics results. The immunohistochemistry methods used in this study are summarized in Appendix A. Immunostaining was performed using a Leica Bond-MAX automation system with Leica Refine detection kits (Leica Biosystems, Bannockburn, IL, USA) following the manufacturer’s instructions. All sections were counterstained with hematoxylin.

### 4.3. Proteomics Analysis Using Mass Spectrometry

We used LMD followed by LC-MS/MS for analyzing CAs in the lung, prostate, and glands of CTAVN. The methods used for LMD with LC-MS/MS have been described previously [57,58].

## Figures and Tables

**Figure 1 ijms-25-04040-f001:**
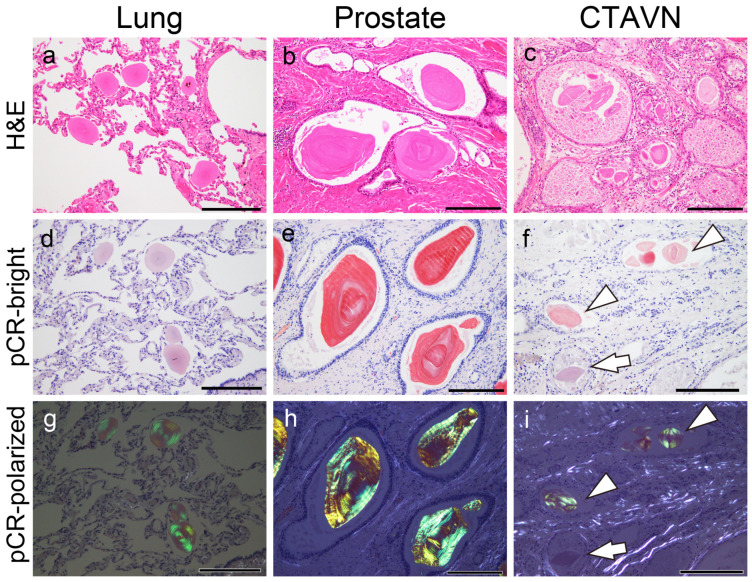
Representative microphotographs of pulmonary and prostatic CAs and CTAVN-associated CAs in the heart. (**a**,**d**,**g**) Pulmonary CA Case 1; (**b**,**e**,**h**) prostatic CA Case 1; (**c**,**f**,**i**) CTAVN Case 1. (**a**–**c**) Hematoxylin and eosin (H&E) staining; (**d**–**f**) phenol Congo red (pCR) staining with a bright field; (**g**–**i**) pCR staining under polarized light. (**a**–**c**) All CAs were round with eosinophilic deposits showing concentric laminations upon H&E staining. Note that macrophages were observed around CAs in the lung (**a**), and multinucleated giant cells were observed around CAs in the prostate (**b**). (**d**,**e**,**g**,**h**) Almost all pulmonary and prostatic CAs exhibited congophilia with apple-green birefringence under polarized light. Although CTAVN-associated CAs showed congophilic deposits (**f**), some CAs (arrow) lacked apple-green birefringence under polarized light (arrowheads indicate CAs showing typical apple-green birefringence). Scale bar = 200 μm (**a**–**i**).

**Figure 2 ijms-25-04040-f002:**
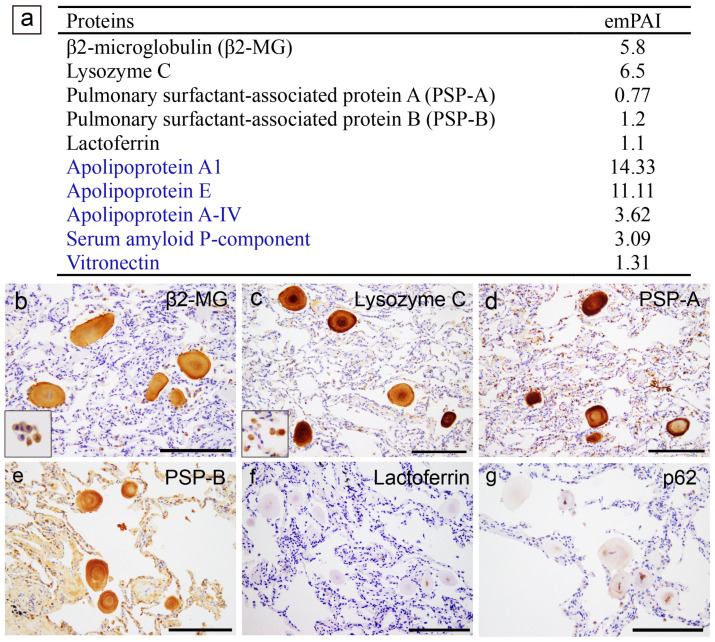
Representative proteomics results and immunohistochemistry micrographs based on these results in the lung. (**a**) Proteins identified in the CAs using laser microdissection and liquid chromatography–tandem mass spectrometry (LMD and LC-MS/MS) in pulmonary CA Case 1; immunohistochemistry for β2-microglobulin (β2-MG) (**b**); lysozyme C (**c**); pulmonary surfactant protein A (PSP-A) (**d**); PSP-B (**e**); lactoferrin (**f**); and p62 (**g**). (**a**) In addition to some amyloid-associated proteins (shown in blue), five amyloidogenic proteins were detected [13]. Of these, immunoreactivity for β2-MG (**b**), lysozyme C (**c**), PSP-A (**d**), and PSP-B (**e**) was confirmed, whereas no immunoreactivity for lactoferrin was observed (**f**). The macrophages were also positive for β2-MG and lysozyme C (insets in panels **b**,**c**). Weak p62 immunoreactivity was observed in the central area of some CAs. The emPAI is the exponentially modified protein abundance index, which is used as an index for estimating relative protein quantification in mass-spectrometry-based proteomic analyses. Scale bar = 200 μm (**b**–**g**).

**Figure 3 ijms-25-04040-f003:**
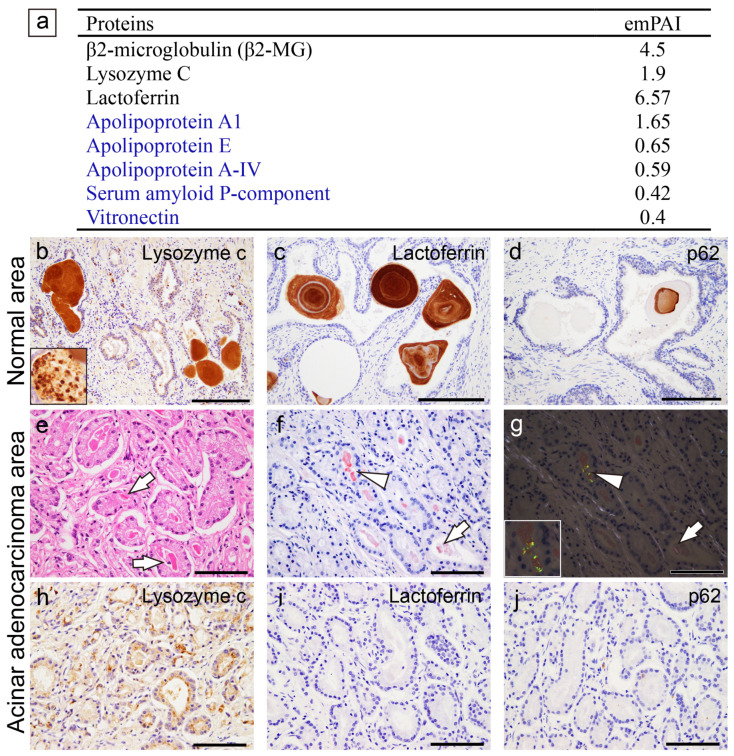
Representative proteomics results and immunohistochemistry micrographs based on the results in the prostate. (**b**–**d**) CAs observed within the normal gland area; (**e**–**j**) eosinophilic deposits observed within the acinar adenocarcinoma glands. (**a**) Proteins identified in the CA lesions using LMD and LC-MS/MS in prostatic CA Case 1; immunohistochemistry for lysosome C (**b**,**h**); lactoferrin (**c**,**i**); p62 (**d**,**j**); H&E staining (**e**); pCR staining under a bright field (**f**) and a polarized field (**g**). (**a**) In addition to some amyloid-associated proteins (shown in blue), three amyloidogenic proteins were detected. VA lesions were positive for lysozyme C (**b**) and lactoferrin (**c**). The macrophages were also positive for lysozyme C (inset in panel **b**). p62 immunoreactivity in the central area of some CAs (**d**). Within the tumor glands, eosinophilic deposits, including crystalloids (arrow), were evident (**e**). (**f**,**g**) Some of the deposits showed weak-to-moderate congophilia and exhibited apple-green birefringence under polarized light (arrowhead, inset is a higher magnification view of the deposit). Crystalloids were weakly positive for pCR but did not exhibit apple-green birefringence under polarized light (arrow). In addition, these eosinophilic deposits were not immunoreactive for lysozyme C (**h**), lactoferrin (**i**), or p62 (**j**). Scale bar = 200 μm (**b**–**d**); 100 μm (**e**–**j**).

**Figure 4 ijms-25-04040-f004:**
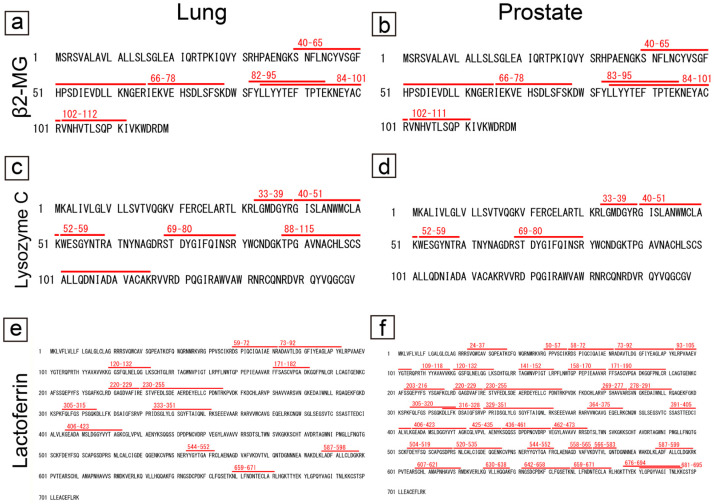
Proteomics results for the amyloidogenic proteins commonly observed in pulmonary and prostatic CAs. (**a**,**c**,**e**) Lung; (**b**,**d**,**f**) prostate; (**a**,**b**) β2-MG; (**c**,**d**) lysozyme C; and (**e**,**f**) lactoferrin. Detected peptides, which had a peptide score of 30 or higher by MASCOT analysis, are shown in red. (**a**–**d**) Regarding β2-MG and lysozyme C, similar peptides were detected in the lung and prostate. (**e**,**f**) In contrast, for lactoferrin, only a few peptide sequences were identified in the lungs, whereas several sequences were identified in the prostate.

**Figure 5 ijms-25-04040-f005:**
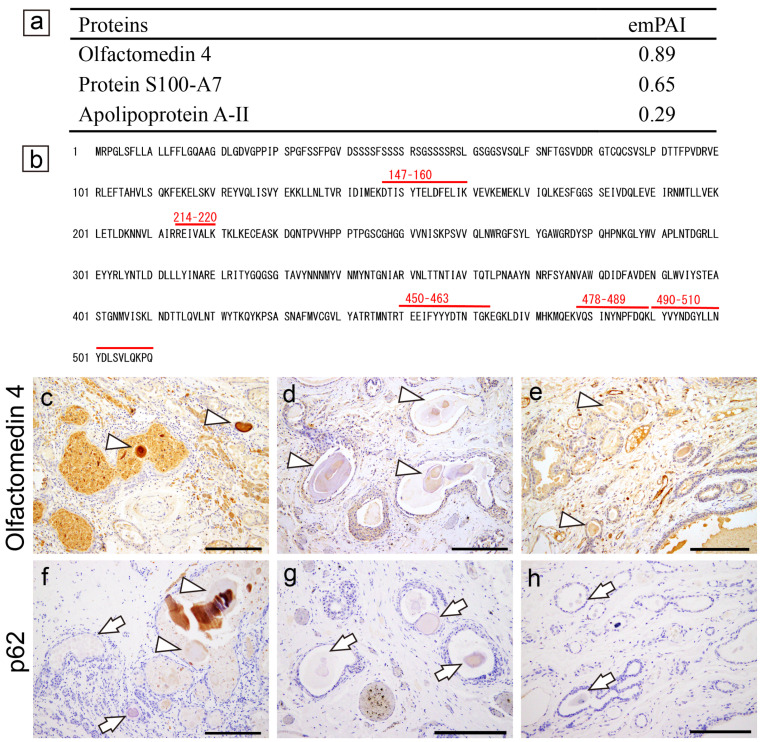
Representative proteomics results and immunohistochemistry micrographs of the heart. (**a**–**c**,**f**) CTAVN Case 1; (**d**,**g**) CTAVN Case 2; (**e**,**h**) CTAVN Case 3. (**a**) Proteins identified in the CAs using LMD and LC-MS/MS in CTAVN Case 1; (**b**) detected peptide sequences of olfactomedin 4; immunohistochemistry for olfactomedin 4 (**c**–**e**); p62 (**f**–**h**). (**a**,**b**) Consistent with our previous immunohistochemistry-based study [4], olfactomedin 4 was identified in CAs and CA-like deposits within CTAVN glands. No amyloid-associated proteins were identified. Detected peptides, which had a peptide score of 30 or higher by MASCOT analysis, are shown in red (**b**). Immunohistochemistry for olfactomedin 4 was strongly positive in Case 1 (**c**) and weakly positive in Case 2 (**d**) and Case 3 (**e**) (arrowheads indicate olfactomedin-4-positive deposits). In contrast, immunohistochemistry for p62 was from negative to partially positive in Case 1 (**f**), whereas it was negative in Cases 2 (**g**) and 3 (**h**) (arrowheads indicate olfactomedin-4-positive deposits and arrows indicate negative deposits). Scale bar = 200 μm (**c**–**h**).

**Figure 6 ijms-25-04040-f006:**
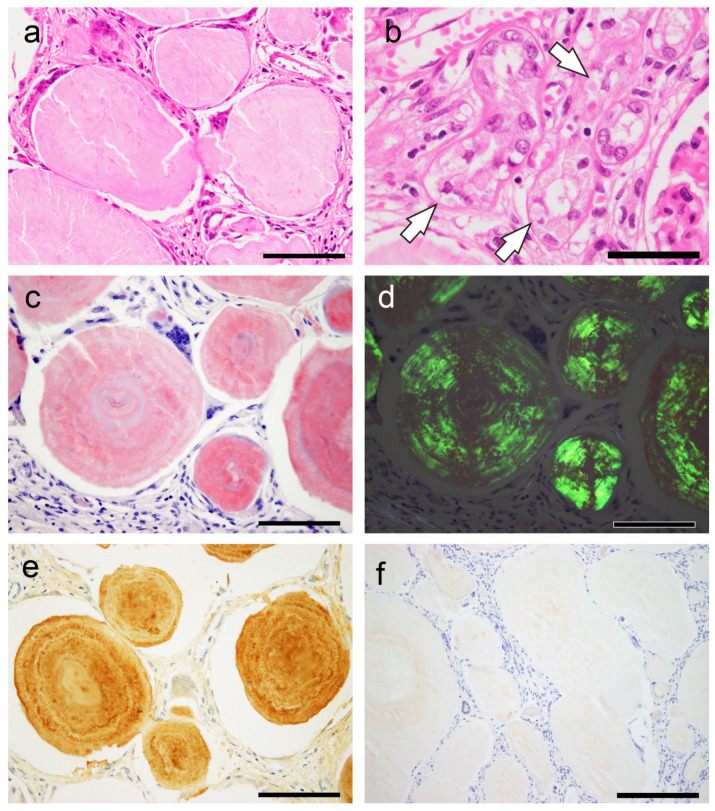
Representative microphotographs of spheroid-type amyloid deposition in the kidney. (**a**,**b**) H&E staining; pCR staining under a bright field (**c**) and polarized light (**d**); immunohistochemistry for Igλ (**e**); p62 (**f**). (**a**,**b**) Spheroid-type amyloid deposits were evident in the interstitium (**a**) and in the tubules (**b**, arrow). (**c**,**d**) These deposits were positive for pCR (**c**) and exhibited apple-green birefringence under polarized light (**d**). (**e**) As shown in our previous report, the deposits were positive by immunohistochemistry for Igλ. (**f**) In contrast, immunohistochemistry for p62 was negative. Scale bar = 200 μm (**f**); 100 μm (**a**–**e**).

**Table 1 ijms-25-04040-t001:** Demographic data associated with cases of corpora amylacea (CA).

	Pulmonary CA Cases			Prostatic CA Cases			CTAVN Cases		
Case #	Case 1 *	Case 2	Case 3	Case 1 *	Case 2	Case 3	Case 1	Case 2	Case 3
Age	75	90	85	75	88	45	36	45	76
Sex	M	F	F	M	M	M	F	M	M
Cause of death	HyT	MN	Drowning	HyT	ACD	KA	SCD	SCD	SCD
Dialysis	None	None	None	None	None	None	None	None	None

Abbreviations: ACD, acute cardiac dysfunction; CTAVN, cystic tumor of the atrioventricular node; F, female; HyT, hypothermia; KA, ketoacidosis; M, male; MN, malnutrition; SCD, sudden cardiac death. * Same patient.

**Table 2 ijms-25-04040-t002:** Summary of previous reports of spheroid-type amyloid deposition (STAD).

Location	S/L	Concomitant Findings	Immunohistochemical Analysis	Proteomic Analysis
Stomach and small intestine [14]	S	Bronchiectasis, FMF, and renal failure	Pos: AA; Neg: β2MG, TTR, Igκ, Igλ, CD68	NE
Small intestine [15,16]	L	Polypoid lesions	Pos: Igλ; Neg: AA, β2MG, TTR, Igκ [15] Pos: AP, AA, Igκ, Igλ (uneven) [16]	NE
Vater ampulla [17]	L	NET	NE	NE
Colon, TI [12]	L	Adenocarcinoma	NE	ALλ
Colon [18,19,20]	L	Ulcerative lesions [18] and rounded lesions [19]	Pos: Igλ [18]	NE
Liver [21,22,23,24,25,26]	S/L	Various diseases (See [21,22])	Pos: AP, AA; Neg: Igκ, Igλ [21] Pos: none; Neg: AP, AA, UB, TTR, Igκ, Ig [23] Pos: AA; Neg: β2MG, TTR, Igκ, Igλ [24] Pos: LECT2 or Ig; Neg: AA, β2MG, TTR [26]	ALECT2 or AL [26]
Sino-nasal tract [27]	L	Nasal mass	Pos: Igκ and Igλ (κ > λ)	NE
Parotid gland [28]	L	Acinic cell carcinoma	NE	NE
URT [29]	S/L	Plasmacytoma	NE	NE
Bronchus [11]	L	Erythematous mass	Pos: Lac; Neg: AA, β2MG, TTR, Igκ, Igλ	ALac
Bone [30,31,32]	L	Myeloma, malignant lymphoma (see [30])	Pos: Igκ or Igλ; Neg: AA [30]	NE
Bone marrow [33]	S	PCP	Pos: Igλ	NE
Ureter [34]	L	Hydronephrosis	NE (likely AA)	NE
Kidney [10]	S	PCP	Pos: Igλ; Neg: AA, β2MG, TTR, Igκ	ALλ
Uterine cervix [35,36,37]	L	Smooth mass [36], SCC [37,38]	Pos: AA; Neg: CK, Igκ, Igλ [35] Pos: CK; Neg: AA, TTR, Igκ, Igλ [36,37]	NE
Pituitary gland [38,39,40,41,42,43,44]	L	Prolactinoma	Pos: PRL; Neg: CK, vimentin, GFAP, GH, FSH, LH, TSH, ACTH, β-A4 [39,41,42]	NE
Breast [45] *	L	Mammary tumor	Pos: α-casein, Lac; Neg: AA, TTR, CK, Igκ, Igλ	NE

Abbreviations: AA, (serum) amyloid A; ACTH, adenocorticotropic hormone; AP, amyloid P component; CK, cytokeratin; FMF, familial Mediterranean fever; FSH, follicle-stimulating hormone; GFAP, glial fibrillary acidic protein; GH, growth hormone; Ig, immunoglobulin (light chain); L, localized; Lac, lactoferrin; LECT2, leukocyte chemotactic factor 2; LH, luteinizing hormone; NE, not evaluated; Neg, negative; NET, neuroendocrine tumor; PCP, plasma cell proliferation; Pos: positive; PRL, prolactin; S, systemic; SCC, squamous cell carcinoma; TI, terminal ileum; TSH, thyroid-stimulating hormone; UB, ubiquitin; URT, upper respiratory tract. * Found in two dogs.

**Table 3 ijms-25-04040-t003:** Summary of the properties of amyloid-forming CAs and STADs.

	Prostatic-CA	Pulmonary-CA	CTAVN-CA	STAD
Congophilia	Strong	Moderate–strong	Weak–moderate	Strong
Strength of the AGBR	Strong	Strong	Weak–moderate	Strong
Macrophages	Positive	Positive	Negative–positive	Positive
Presence of CPs	Positive	Positive	Negative	Negative
Presence of AAPs	Positive	Positive	Negative	Positive
p62-IR	Positive (focal)	Positive (focal)	Negative–positive (focal)	Negative

Abbreviations: AAPs, amyloid-associated proteins; AGBR, apple-green birefringence under polarized light; CPs, common proteins (β2-microglobulin and lysozyme C); IR, immunoreactivity.

## Data Availability

The datasets used and analyzed in the current study are available from the corresponding authors upon request.

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
