# Peer review of "Amyloid-Forming Corpora Amylacea and Spheroid-Type Amyloid Deposition: Comprehensive Analysis Using Immunohistochemistry, Proteomics, and a Literature Review"

_ijms, 2024, doi:10.3390/ijms25074040_

Round 1
Reviewer 1 Report
Comments and Suggestions for Authors
The manuscript comprehensively compares and summarizes various methods of the examination of amyloid-forming corpora amylacea and spheroid-type amyloid deposition, elucidating their similarities and differences.
In general, the manuscript is well-written and easy to follow, the presented data are interesting and useful. However, in my opinion the value of the paper can be improved.
Could the authors consider carrying out additional experiments focused on detailed examination of structural characteristics of amyloid structures within deposits? This may contribute to an increase of depth insight of similarities and differences between amyloid-forming corpora amylacea (CA) in the prostate and lung.
Author Response
We thank the Reviewer for their kind review and comments, which helped us substantially improve our manuscript. In the revised manuscript, the newly added statements are indicated in red font.
The manuscript comprehensively compares and summarizes various methods of the examination of amyloid-forming corpora amylacea and spheroid-type amyloid deposition, elucidating their similarities and differences. In general, the manuscript is well-written and easy to follow, the presented data are interesting and useful. However, in my opinion the value of the paper can be improved. Could the authors consider carrying out additional experiments focused on detailed examination of structural characteristics of amyloid structures within deposits? This may contribute to an increase of depth insight of similarities and differences between amyloid-forming corpora amylacea (CA) in the prostate and lung.
Response:
We thank the Reviewer for their salient suggestion. We wholeheartedly agree that further morphological examinations, such as cryogenic electron microscopy, would improve the quality of our paper. However, this additional examination did not meet the deadline for the revised manuscript. Therefore, we have added this as a limitation of our study (lines 326–332).
Reviewer 2 Report
Comments and Suggestions for Authors
The study aims to compare amyloid-forming corpora amylacea (CA) in the prostate and lung tissues and examine CA in cystic tumors of the atrioventricular node (CTAVN). Additionally, it seeks to distinguish between amyloid-forming CA and spheroid-type amyloid deposition. The comments provided will be addressed in the study. However, I would like to offer the following suggestions for enhancements that could be implemented:
- Can you discuss any potential challenges or limitations related to sample size (such as the risk of findings not accurately reflecting the broader population, thus limiting the applicability of the results), sampling bias (for instance, focusing solely on specific demographic groups or geographic regions), and/or confounding variables?
- What potential areas of future research do you envision for amyloid-forming corpora amylacea (CA) and spheroid-type amyloid deposits? Additionally, how could these research directions be effectively incorporated into clinical practice?
- Could you elaborate on the role of gender, age, or genetic factors in amyloid-forming corpora amylacea (CA) and spheroid-type amyloid deposits and the potential impact on the effectiveness of therapeutic strategies?
- What scale is used for histopathological images? could you make a score for the observed data?
Author Response
Response to Reviewer #2
We thank the Reviewer for their kind review and comments, which have helped us substantially improve our manuscript. In the revised manuscript, the newly added statements are indicated in red font.
The study aims to compare amyloid-forming corpora amylacea (CA) in the prostate and lung tissues and examine CA in cystic tumors of the atrioventricular node (CTAVN). Additionally, it seeks to distinguish between amyloid-forming CA and spheroid-type amyloid deposition. The comments provided will be addressed in the study. However, I would like to offer the following suggestions for enhancements that could be implemented:
- Can you discuss any potential challenges or limitations related to sample size (such as the risk of findings not accurately reflecting the broader population, thus limiting the applicability of the results), sampling bias (for instance, focusing solely on specific demographic groups or geographic regions), and/or confounding variables?
Response:
We thank the Reviewer for their insightful query. Accordingly, we have clarified the limitations of our study in the revised Discussion section (lines 326–332).
- What potential areas of future research do you envision for amyloid-forming corpora amylacea (CA) and spheroid-type amyloid deposits? Additionally, how could these research directions be effectively incorporated into clinical practice?
Response:
We thank the Reviewer for their important question. Future studies should elucidate the differences in the amyloid fiber structure between CA and STAD using cryogenic electron microscopy. We have added this explanation in lines 329–332. We believe that the elucidation of the mechanisms shared between CA and STAD can contribute to the development of therapies that stimulate amyloid elimination via mechanisms that differ from those based on antibodies. We have added this consideration in lines 349–357.
- Could you elaborate on the role of gender, age, or genetic factors in amyloid-forming corpora amylacea (CA) and spheroid-type amyloid deposits and the potential impact on the effectiveness of therapeutic strategies?
Response:
We thank the Reviewer for their insightful comments and questions. The CA frequency increases with age, suggesting that age-related changes in protein production and efflux impact CA formation. As many amyloidoses are associated with aging, age-related changes in protein metabolism may be important factors underlying pathogenesis. Therefore, we suggest that CA analysis can identify specific protein metabolic mechanisms that are altered with aging and offer insights into approaches that can be utilized to prevent amyloid deposition. We have added this consideration in lines 351–357. Regarding sex and genetic factors, the frequency of ATTR amyloidosis, in particular, varies by region, and males are more prone to severe disease [Amyloid 2012; 19(S1): 68–70]. Thus, we consider that these factors are also important. However, their impact on CA and STAD is currently unknown.
- What scale is used for histopathological images? could you make a score for the observed data?
Response:
We thank the Reviewer for their question and apologize for the lack of clarity regarding the scale bars. Scale bars were created based on data measured using a microscope camera at the time the images were acquired. The sizes of the scale bars can be found at the end of the figure legend.